# Investigation of an EHV-1 Outbreak in the United States Caused by a New H_752_ Genotype

**DOI:** 10.3390/pathogens10060747

**Published:** 2021-06-13

**Authors:** Nicola Pusterla, Samantha Barnum, Julia Miller, Sarah Varnell, Barbara Dallap-Schaer, Helen Aceto, Aliza Simeone

**Affiliations:** 1Department of Veterinary Medicine and Epidemiology, School of Veterinary Medicine, University of California, Davis, CA 95616, USA; smmapes@ucdavis.edu; 2Alliance Equine Health Care, Glenmoore, PA 19343, USA; miller.julia.vmd@gmail.com (J.M.); svarnelldvm@gmail.com (S.V.); 3Department of Clinical Studies, New Bolton Center, University of Pennsylvania, Kennett Square, PA 19348, USA; bldallap@vet.upenn.edu (B.D.-S.); helenwa@vet.upenn.edu (H.A.); 4Pennsylvania Department of Agriculture, Bureau of Animal Health, Collegeville, PA 17110, USA; asimeone@pa.gov

**Keywords:** EHV-1, new genotype, H752, outbreak, horses

## Abstract

Here we report on an EHV-1 outbreak investigation caused by a novel genotype H_752_ (histidine in amino acid position 752 of the *ORF 30* gene). The outbreak involved 31 performance horses. Horses were monitored over a period of 35 days for clinical signs, therapeutic outcome and qPCR results of EHV-1 in blood and nasal secretions. The morbidity of the EHV-1 outbreak was 84% with 26 clinically infected horses displaying fever and less frequently anorexia and distal limb edema. Four horses showed mild transient neurological deficits. Clinically diseased horses experienced high viral load of EHV-1 in blood and/or nasal secretions via qPCR, while subclinically infected horses had detectable EHV-1 mainly in nasal secretions. The majority of infected horses showed a rise in antibody titers to EHV-1 during the outbreak. All 31 horses were treated with valacyclovir, while clinically infected horses further received flunixin meglumine and sodium heparin. This investigation highlights various relevant aspects of an EHV-1 outbreak caused by a new H_752_ genotype: (i) importance of early detection of EHV-1 infection; (ii) diagnostic challenge to assess H_752_ genotype; (iii) apparent benefit of valacyclovir use in the early stage of the outbreak; and (iv) weekly testing of blood and nasal secretions by qPCR in order to monitor individual infection status and lift quarantine.

## 1. Introduction

Equine herpesvirus-1 (EHV-1) is an important, ubiquitous equine viral pathogen that exerts its major impact by inducing respiratory disease in young horses, sporadic abortions or abortion storms in pregnant mares, early neonatal death in foals, and myeloencephalopathy [1]. Although equine herpesvirus-1 myeloencephalopathy (EHM) is a relatively uncommon manifestation of EHV-1 infection, it can cause devastating losses and severely impact the equine industry, as exemplified by recent outbreaks at riding schools, racetracks, horse shows and veterinary hospitals throughout North America, Europe and New Zealand [2,3,4,5,6].

The increasing application of qPCR for the molecular detection of EHV-1 in practice settings has presented new dilemmas with regard to how test results are interpreted and used by both equine practitioners and regulatory veterinarians. A recently identified single nucleotide polymorphism (SNP) at position 2254 in the DNA polymerase gene (*ORF 30*) has been shown to correlate with neurological disease [7,8]. This SNP is responsible for a single amino acid residue at position 752 of the DNA polymerase with EHV-1 strains associated with neurological outbreaks involving a D_752_ genotype (referred to as G_2254_), whereas most non-neurological outbreaks involve a N_752_ genotype (referred to as A_2254_). Strain characterization may be important given that the potential of EHM development is greater in horses infected with a neuropathogenic genotype (D_752_). Furthermore, detection of a neuropathogenic EHV-1 strain may influence therapy, especially the use of antiviral drugs such as valacyclovir, used to decrease viremia and prevent the development of neurological sequelae. Since asparagine (N) and aspartic acid (D) are considered the only two amino acids identified at position 752 of the DNA polymerase gene of EHV-1 [7], the majority of diagnostic laboratories in the USA are often only using allelic discrimination qPCR assays to detect EHV-1. Unfortunately, such a molecular approach may lead to missed EHV-1 cases as exemplified by the present outbreak and by a recently reported outbreak from France [9].

Here, we report on an EHV-1 outbreak investigation caused by a novel genotype H_752_. The outbreak involved 31 mostly adult performance horses. Horses were monitored over a period of 35 days for clinical signs, therapeutic interventions and qPCR results of EHV-1 in blood and nasal secretions. Furthermore, serology was performed at the onset of the outbreak and during the convalescent period. This outbreak features unique observations, including the importance of early detection of EHV-1 infection, the diagnostic challenge to detect the new H_752_ genotype, the apparent benefit of using valacyclovir in the early stage of the outbreak in order to prevent severe myeloencephalopathy and the regular testing of blood and nasal secretions by qPCR in order to monitor individual infection status and lift quarantine.

## 2. Results

On 9 March 2021, two of the authors (JM and SV) visited a 31-horse show barn in order to perform routine dental care and vaccination. As part of the health assessment of every horse, a physical examination was performed on various horses and 10 horses were unexpectedly displaying elevated rectal temperature (38.9 to 40.2 °C, median 39.2 °C). At that time, all horses appeared bright and alert. The attending veterinarians collected whole blood and nasal secretions from 10 febrile horses for the detection of common respiratory viruses (equine influenza virus, equine rhinitis A and B virus, equine herpesvirus-1 and -4) using quantitative (q)PCR. Eight out of the 10 horses tested qPCR-positive in blood and/or nasal secretions for EHV-1 targeting the *gB* gene, while the *ORF 30* genotype assays were unable to differentiate between D_752_ and N_752_. The unusual molecular results and lack of systemic clinical signs at the exception of fever prompted the investigation of this outbreak. In the following 35 days, each of the 31 horses was monitored daily for abnormal physiological parameters, blood and nasal secretions were collected once weekly for the qPCR detection of EHV-1 and blood was collected 35 days apart to measure specific antibodies to EHV-1.

The horse population was composed of 14 geldings and 17 mares, aged 1 to 27 years (median 4 years). Various breeds were represented, including Warmblood (24 horses), Welsh pony (4), Thoroughbred (2), and Thoroughbred/draft horse cross (1). Twenty-seven horses were kept in the main barn and 4 horses were stabled in a shed row barn located 30 m from the entrance to the main barn. 

Throughout the 35-day observation period, 26/31 horses developed fever with peak temperatures ranging from 38.7 °C to 41.0 °C (median 39.3 °C, Table 1). The fever lasted from 1 to 8 days (median 2.5 days). Additional clinical signs reported included anorexia (7 horses), distal limb edema (3), and mild neurological deficits such as ataxia, weakness and proprioceptive deficits (4). Twenty-one horses developed abnormal clinical signs during the first week of the outbreak, while 2 and 3 horses developed abnormal clinical signs during the second and third week of the outbreak, respectively. Five horses did not develop any abnormal clinical signs during the entire observation period. 

On the first sample collection (Day 0), 23 and 13 horses tested qPCR-positive for EHV-1 in nasal secretions and blood, respectively (Table 2). The EHV-1 viral loads at the gDNA level in nasal secretions ranged from 412 to 1.78 × 10^8^ gene copies/million cells (median 6.3 × 10^5^ gene copies/million cells) and the viral load in blood ranged from 572 to 2.1 × 10^5^ gene copies/million cells (median 6.2 × 10^3^ gene copies/million cells, Figure 1). On the second collection time point (Day 14), 14 horses tested qPCR-positive for EHV-1 in nasal secretions, while all blood samples tested negative. The EHV-1 viral loads in nasal secretions ranged from 105 to 2.34 × 10^7^ gene copies/million cells (median 3203 gene copies/million cells). On day 14, 10 and 2 horses tested EHV-1 qPCR-positive in nasal secretions and blood respectively. The EHV-1 viral loads in nasal secretions ranged from 94 to 4.10 × 10^4^ gene copies/million cells (median 1204 gene copies/million cells). The two horses with EHV-1 qPCR-positive blood samples yielded viral loads of 2328 and 3987 gene copies/million cells. On day 21, five and two horses tested qPCR positive for EHV-1 in nasal secretions and blood respectively. The EHV-1 viral load in nasal secretions ranged from 245 to 2.30 × 10^5^ gene copies/million (median 800 gene copies/million cells). The two horses with EHV-1 qPCR-positive blood samples yielded viral loads of 127 and 311 gene copies/million cells. On day 28, five horses tested EHV-1 qPCR-positive in nasal secretions while all horses tested negative in blood. The EHV-1 viral load in nasal secretions ranged from 169 to 1636 gene copies/million (median 241 gene copies/million cells). By day 35, all horses tested EHV-1 qPCR-negative in nasal secretions and blood. 

Treatment of the study horses consisted of the administration of valacyclovir (Camber Pharmaceuticals, Piscataway, NJ, USA) in all horses, and flunixin meglumine (Merck Animal Health, Madison, NJ, USA) and sodium heparin (Sagent Pharmaceuticals, Schaumburg, IL, USA) in all febrile horses (Table 3). All horses treated with valacyclovir experienced a strong decline in EHV-1 viral load in nasal secretions and all horses had no detectable EHV-1 by qPCR in blood on day 7. However, susceptibility to EHV-1 in horses remained, as exemplified by five horses that showed clinical and molecular evidence of EHV-1 infection after the valacyclovir treatment was discontinued.

Serology performed at onset (day 0) of the outbreak and 35 days later showed that 7 and 23 horses had a positive titer on day 0 and day 35, respectively (Table 1). Seroconversion, defined as an increase in OD of >0.129 above the previous absorbance reading, was observed in 16 horses (Figure 2). Two horses with doubtful results on day 0 became positive on day 35, and two seronegative horses on day 0 became doubtful on day 35. Six horses had no detectable antibodies at the two time points. Among the latter six horses, five horses had clinical signs and molecular detection of EHV-1, while one horse had no evidence of infection.

qPCR positive samples from five individual horses (three febrile horses from the performance horse barn and two neurological horses temporally and epidemiologically associated with the outbreak but from different locations) were further sequenced to determine the partial nucleotide and amino acid sequence of the *ORF 30* gene. Sequencing was performed since all EHV-1 qPCR positive horses tested only positive when targeting the *gB* gene and could not be further characterized into either a D_752_ or a N_752_ genotype. The 663 bp product of the *ORF 30* gene showed 100% homology among the five individual horses. A single nucleotide polymorphism, cytosine, was found at position 2254, translating into histidine at position 752 of the amino acid sequence (H_752_). 

The origin of the outbreak could not be determined with certainty. However, one month prior to the onset of the outbreak, a mare was referred to a local hospital for an elective procedure. The mare returned to the performance horse barn, where she stayed for 2 weeks before leaving for another barn. On the two-week recheck (end of February), the mare appeared in good health and displayed normal physical findings. In early March, the local hospital reported on a hospitalized horse that was diagnosed with EHM and was euthanized within hours of admission. The diagnosis of EHM was further confirmed on necropsy. DNA of this horse was available for the *ORF 30* gene sequencing. Two days later, one additional horse, stabled in the adjacent state of Maryland, was diagnosed with EHM^.^. This horse was already recumbent when seen on emergency by the ambulatory service and was euthanized. It had direct contact to a horse that came back from the local hospital about 3 weeks earlier, but had exhibited hind limb edema and discomfort thought to be laminitis-related about 1 week prior to becoming acutely recumbent. DNA from the second EHM case was also available for the *ORF 30* gene sequencing. The horse in contact with the second EHM case was hospitalized during the risk period identified by epidemiological trace-back and trace-forward, during which both the mare from the performance barn and another horse subsequently identified as EHV-1 qPCR-positive were present in the hospital. The latter was still hospitalized when the first EHM case was originally in the hospital for an unrelated reason about 10 days prior to presenting with EHM. No additional EHV-1 cases were diagnosed in the hospital after the EHM case. The quarantine at the performance horse barn was lifted following the last negative EHV-1 qPCR results on day 35. According to the farm manager and attending veterinarians, no additional cases have been diagnosed since the quarantine was lifted. 

## 3. Discussion

One of the unique features of this EHV-1 outbreak is that the investigation started when horses did not display apparent clinical signs. The fact that the attending veterinarians noticed that the majority of the performance horses displayed elevated rectal temperature without additional clinical signs emphasizes the importance of regularly assessing the rectal temperature of at-risk horses. One could argue that without early detection of EHV-1 and medical intervention, more horses might have developed myeloencephalopathy, considering the neuropathogenic potential of the H_752_ genotype strain. From an epidemiological standpoint, the molecular signature of this strain allowed to reconstruct possible transmission paths between a local veterinary hospital and the performance horse barn. Another relevant observation is the mutation of the EHV-1 strain at the pivotal nucleotide position 2254, which is the key position previously shown to discriminate between neuropathogenic (D_752_) and non-neuropathogenic (N_752_) genotypes [7,10]. Since the full genome sequencing of two prototype EHV-1 strains, Ab4 (neuropathogenic) and V592 (non-neuropathogenic), the majority of diagnostic laboratories in the USA rely solely on allelic discrimination qPCR assays to detect EHV-1 [11,12,13,14]. Unfortunately, such a molecular approach may lead to missed EHV-1 cases as exemplified by the present outbreak and by a recently reported outbreak from France [9].

The genetic characterization of the EHV-1 strain showed a new C_2254_ (H_752_) mutation of the *ORF 30* gene, similar to the mutation reported in a recent outbreak from France [9]. The clinical signs reported from the French horses were respiratory signs (96.9%), pyrexia (70.3%), limb edema (59.4%) and lethargy (28.1%) [9]. In the performance horses from Pennsylvania, pyrexia (84%) was the predominant clinical sign. The lack of respiratory signs seen in the performance horses might be explained by the recent vaccination of the horse population against the respiratory form of EHV-1. All the performance horses were on a bi-annual vaccination schedule for equine influenza and rhinopneumonitis and time from last vaccine to onset of outbreak was 1 to 6 months with a median of 1 month. Both the H_752_ EHV-1 strain from France and Pennsylvania showed neuropathogenicity. Among the French horse population, two horses showed neurological signs of disease, which required euthanasia in one of them [9]. In the present outbreak, two adult horses outside the show barn had to be euthanized due to severe neurological deficits. It is of interest to notice that 4 out of the 31 performance horses developed mild and transient neurological deficits early in the disease. It is likely that early medical intervention with valacyclovir in all performance horses may have prevented severe neurological sequelae. The use of valacyclovir has been shown to significantly decrease viral replication and signs of disease in EHV-1-infected horses, especially when treatment is initiated before the onset of neurological deficits [15]. The anti-viral effect of valacyclovir was evident in the study horses, as viremia decreased from 42% at onset of treatment to 0% within 7 days. Valacyclovir also reduced the number of horses shedding EHV-1 as well as the amount of virus shed within the first 7 days of treatment. Previous work on valacyclovir has shown that replication stops immediately after initiation of treatment for alpha herpesviruses [16]. The present data are in agreement with a previous study showing the rapid decline in viral loads in blood and nasal secretions from horses with EHM treated with valacyclovir [17]. However, the susceptibility to EHV-1 infection was retained following discontinuation of valacyclovir, as exemplified by five horses that developed abnormal clinical signs during the second or third week of the outbreak. The medical management of EHV-1 outbreaks with anti-herpetic, anti-inflammatory and anti-thrombotic drugs is generally aimed at reducing the risk of EHM by either reducing or preventing viremia and by mitigating the interactions of EHV-1 with endothelial cells [18,19]. Unfortunately, there are too few placebo-controlled studies to evaluate the efficacy of such medical protocols in the prevention of EHM. 

Until recently, asparagine (N) and aspartic acid (D) have been the only two amino acids identified at position 752 of the DNA polymerase gene of EHV-1 [7]. In 2020, a French group reported on the fortuitous identification and characterization of a new ORF 30 genotype H_752_ associated with one specific outbreak that lasted several weeks in 2018 [9]. Similar to the Pennsylvania outbreak, the molecular profile of the EHV-1 strain, i.e., positive by a conventional qPCR *gB* method and negative by a qPCR *ORF30* N/D_752_ genotyping method, prompted further investigations. From an evolutionary standpoint, it appears that the H_752_ genotype possibly evolved from the N_752_ genotype [9]. The discovery of a new genotype H_752_ adds to the already complex pathophysiology and diagnostic challenges of EHV-1 [20]. The French and Pennsylvanian strains of EHV-1, while mostly causing a self-limiting disease characterized by fever, lethargy, anorexia, respiratory signs and distal limb edema, has also shown neuropathogenic potential. The distribution of the new H_752_ genotype in horse populations is yet unknown, mostly because most diagnostic laboratories in the USA base the EHV-1 detection solely on allelic discrimination of the *ORF 30* N/D_752_ genotyping. Future studies aimed at studying the frequency of the new H_752_ genotype using a multi-gene assay approach are urgently needed. 

Recent vaccination against EHV-1 was unable to prevent the development of clinical disease with the H_752_ genotype. It is, however, possible that the recent vaccination in most of the performance horses may have reduced disease severity. While the majority of the performance horses seroconverted or showed a significant increase in OD, six horses remained seronegative. One seronegative horse remained uninfected, while the remaining five horses had clinical signs and molecular detection of EHV-1. For the latter five horses, it is very likely that the early use of valacyclovir may have negatively impacted the antibody production through rapid elimination of the virus. The failure for most EHV-1 vaccines to confer protection against natural infection likely relates to the immunomodulatory properties of EHV-1, similar to what is known for other herpesviruses [21,22].

The original index case was likely a subclinically infected horse. Movement of horses between show barns and a local veterinary hospital contributed to the spread of EHV-1 and highlight the risk of viral spread even in the absence of clinical disease. The study also showed the high contagiousness of the new H_752_ variant with a morbidity of 84% and an overall infection rate of 97%. While the spread of equine alpha herpesviruses is almost impossible to prevent at large equine operations, the daily monitoring of at-risk horses and use of proper biosecurity protocols around horses undergoing transportation may reduce the risk of EHV-1 spread. 

## 4. Materials and Methods

### 4.1. Horses

The outbreak was first recognized on 9 March, when various performance horses were presented to two attending veterinarians for routine dental care and vaccination. The barn, located in rural south-eastern Pennsylvania, had 31 horses present at the onset of the outbreak. Clinical data and individual medical treatments were recorded in real time from the onset of the outbreak (day 0) up to the lifting of the quarantine (day 35). Sample collection was performed under the horse owners’ consent.

### 4.2. Quantitative Polymerase Chain Reaction (qPCR) for Detection of EHV-1

Whole blood was collected in evacuated tubes (BD Vacutainer^®^, Franklin Lakes, NJ, USA) and nasal secretions were collected using 6-inch-long rayon-tipped swabs (Puritan^®^ Sterile Rayon Tipped Applicators, Guilford, ME, USA). Nucleic acid extraction from whole blood and nasal secretions was performed 24 h post-collection using an automated nucleic acid extraction system (QIAcube HT, Qiagen, Valencia, CA, USA) according to the manufacturer’s recommendations. All samples were assayed for the presence of the *equine glyceraldehyde-3-phosphate dehydrogenase* (*eGAPDH*) gene, the *glycoprotein B* (*gB*) and the *polymerase* (*ORF 30*) gene of EHV-1 using previously reported real-time TaqMan PCR assays [23].

### 4.3. Serology

A commercially available type-specific ELISA (Svanovir™, Svanova Biotech AB, Uppsala, Sweden) was used to determine EHV-1 antibody concentrations in serum from the study horses. Based upon the manufacturer’s recommendations, antibody concentrations were considered negative, doubtful or positive if the optical density (OD) of the test well was <0.1, 0.1–0.2 and >0.2 OD respectively. Seroconversion in horses was defined as an increase in OD of >0.129 above the previous absorbance reading, as previously described for this assay [24].

### 4.4. Sequencing of an ORF 30 Gene Fragment

A 663-bp product of the *ORF 30* gene was sequenced in order to determine any possible nucleotide mismatch with the preexisting genotype assays at position 2254. PCR primers used for the amplification were forward primer 1901F-TTGGGCACAGTCAGGCAG and reverse primer 2564R-TACGTTCACTCTCGTTGGGC. Amplification of the partial *ORF 30* gene was performed using Advantage 2 Polymerase chemistry (Takara Bio, Mountain View, CA, USA). The amplification conditions were as follows: 4 min at 95 °C, and 40 cycles of 1 min at 95 °C, 1 min at 56 °C and 1 min at 72 °C. The final elongation step included 10 min at 72 °C. Sequencing products were purified using QIAquick PCR purification kit (Qiagen, Germantown, MD, USA). Sequencing of the purified PCR products was performed by the UCDNA sequencing Facility (Davis, CA, USA). Sequences were analyzed using SeqScanner (Applied Biosystems, Foster City, CA, USA) and Primer Express (Applied Biosystems). 

## 5. Conclusions

This study describes the investigation of an EHV-1 outbreak in a group of 31 performance horses caused by a new 2254 mutant strain (C_2254_/H_752_). The outbreak was associated with fever and occasional neurological deficits. The diagnostic challenge when dealing with this new H_752_ genotype is that most diagnostic laboratory solely rely on allelic discrimination (D_752_ versus N_752_) qPCR assays to detect EHV-1. It is, therefore, recommended to use molecular laboratories that either incorporate a universal EHV-1 target gene or expand their ORF 30 SNP assays to include the new H_752_ genotype. Close clinical monitoring of at-risk horses has remained the main strategy in disease prevention. The detection of EHV-1 early in the outbreak allowed medical intervention with the goal to prevent myeloencephalopathy. Specifically, the use of valacyclovir in all horses was successful at reducing and preventing viremia and nasal shedding. Furthermore, regular testing of blood and nasal secretions by qPCR allowed to closely monitor individual infection status and document the resolution of the outbreak. 

## Figures and Tables

**Figure 1 pathogens-10-00747-f001:**
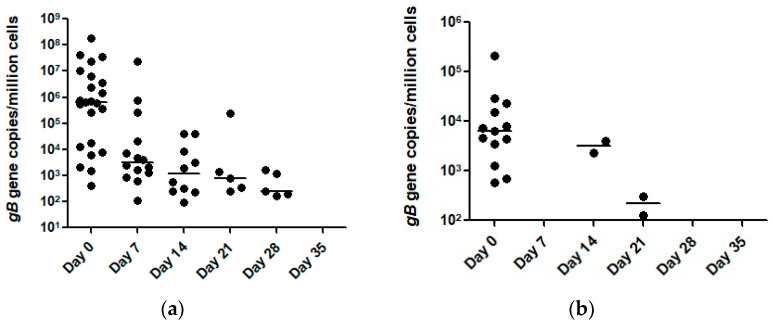
Temporal qPCR results of EHV-1 detection in nasal secretions (**a**) and blood (**b**) from 31 horses involved in an EHV-1 outbreak. The horizontal bars represent the median number of EHV-1 *gB* target genes/million cells.

**Figure 2 pathogens-10-00747-f002:**
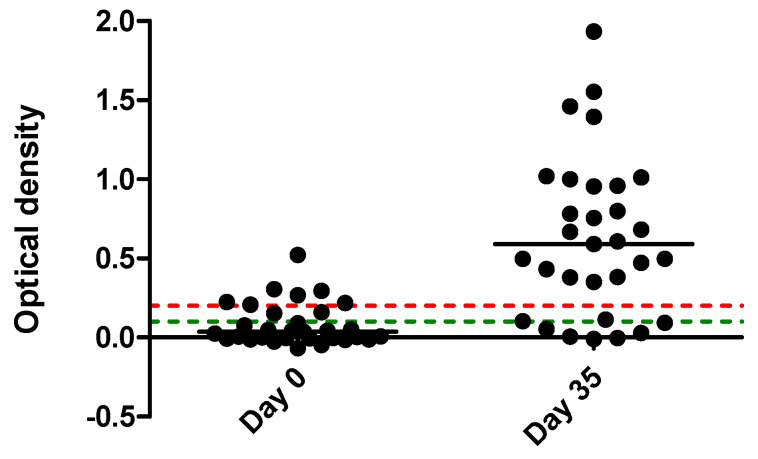
Titers, expressed as optical density on day 0 and 35 from 31 horses involved in an EHV-1 outbreak. Based upon the manufacturer’s recommendations, antibody concentrations were considered negative, doubtful or positive if the optical density (OD) of the test well was <0.1, 0.1–0.2 and >0.2 OD respectively. The green and red dotted lines represent cutoffs at 0.1 and 0.2 OD, respectively.

**Table 1 pathogens-10-00747-t001:** Clinical, molecular and serological results from 31 horses involved in an EHV-1 outbreak.

Horse	Clinical Signs		qPCR		Serology	
	Fever in °C (Peak/Duration)	Other Signs	Nasal Shedding	Viremia	Day 0	Day 35
1	38.8/6	-	-	-	-	-
2	39.3/2	DLE	+	+	-	-
3	No	-	+	+	+	+
4	40.6/3	-	+	+	doubtful	+
5	39.7/3	-	+	+	-	+
6	39.9/4	-	+	+	+	+
7	39.6/6	-	+	-	+	+
8	39.3/1	-	+	-	-	+
9	38.9/1	-	+	-	-	+
10	40.1/2	A	-	+	-	doubtful
11	38.8/1	A	+	+	-	-
12	38.9/1	-	+	+	-	+
13	40.1/3	A	+	+	-	+
14	38.7/2	-	+	-	-	+
15	39.2/1	-	+	-	-	-
16	38.7/1	-	+	-	-	-
17	38.8/1	-	+	-	+	+
18	40.9/5	ND	+	+	-	+
19	41.0/5	A	+	+	-	+
20	40.5/8	A	+	-	-	+
21	No	-	+	-	-	+
22	39.7/4	A,ND	+	+	-	+
23	40.1/5	-	+	+	+	+
24	39.4/2	ND	-	+	doubtful	+
25	No	-	+	-	+	+
26	39.1/4	-	-	+	-	+
27	39.3/4	-	+	+	+	+
28	40.6/6	DLE	+	+	-	+
29	No	-	+	-	-	+
30	40.9/2	A,DLE,ND	-	+	-	doubtful
31	No	-	-	-	-	-
Total	Fever (26 horses) No fever (5)	A (7) DLE (3) ND (4)	negative (5) positive (26)	negative (13) positive (18)	negative (22) doubtful (2) positive (7)	negative (6) doubtful (2) positive (23)

A = anorexia; DLE = distal limb edema; ND = neurological deficits (ataxia, weakness, proprioceptive deficits).

**Table 2 pathogens-10-00747-t002:** EHV-1 qPCR viral load results in nasal secretions and blood from 31 horses involved in an EHV-1 outbreak.

Day	qPCR Results of Nasal Secretions		qPCR Results of Blood	
	pos/neg	Range (Median)	pos/neg	Range (Median)
0	23/8	412–1.78 × 10^8^ (6.3 × 10^5^)	13/18	572–2.1 × 10^5^ (6269)
7	14/17	105–2.34 × 10^7^ (3203)	0/31	0
14	10/21	94–4.10 × 10^4^ (1204)	2/29	2328–3987 (3158)
21	5/26	245–2.30 × 10^5^ (800)	2/29	127–311 (219)
28	5/26	169–1636 (241)	0/31	0
35	0/31	0	0/31	0

The range and median are expressed as number of EHV-1 *gB* target genes per million cells.

**Table 3 pathogens-10-00747-t003:** Treatment administered to 31 horses involved in an EHV-1 outbreak.

Drug	Dose	Treatment Days (Median)	Horses Treated
Valacyclovir	Loading 30 mg/kg q8h PO for first 6 doses Maintenance 20 mg/kg q12h PO	11–25 (12)	31
Flunixin meglumine	0.5–1.1 mg/kg q12h to q24h PO or IV	1–20 (4)	26
Sodium heparin	50 IU/kg q12h SQ	1–3 (3)	26

PO = via oral route, IV = via intravenous route, SQ = via subcutaneous route.

## Data Availability

Data available on request due to privacy restrictions.

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
