# Peer review of "Investigation of an EHV-1 Outbreak in the United States Caused by a New H752 Genotype"

_pathogens, 2021, doi:10.3390/pathogens10060747_

Round 1

Reviewer 1 Report

This is a well constructed investigation, and the manuscript is lucid and well written.  A few minor points are offered by the reviewer.

Author Response

Dear Reviewer,

We herewith submit the revisions to our manuscript entitled “Investigation of an EHV-1 Outbreak in the United States Caused by a new H752 Genotype” by Nicola Pusterla, Samantha Barnum, Julia Miller, Sarah Varnell, Barbara Dallap-Schaer , Helen Aceto and Aliza Simeon. We appreciate the time and comments provided by the reviewer, which we agree greatly improve the manuscript. We have addressed the suggested changes to the manuscript, and hope that our manuscript is now acceptable for publication.

Thank you for your consideration and we look forward to hearing back from you at your earliest convenience.

Sincerely,

Nicola Pusterla

Comments to Reviewer 1

  1. It would be helpful to the non-EHV-1 expert if the exact definition of the H752 genotype were offered in the Abstract.

Additional information pertaining to the new genotype has been added in the abstract.

Also, it is not perfectly clear if this report is the first detection of the  ‘new” H752 genotype of EHV-1.  This is the first report of this new genotype in the USA.

  1. The title of the paper states that this is an “atypical” EHV-1 outbreak. It would be helpful for the average reader who is likely not an expert on EHV-1 clinical diseases to explain the “atypical” aspects of this outbreak.

Atypical refers more to the initial clinical presentation as the outbreak was detected while the horses did not appear systemically ill. In order to prevent any confusion to the reader, the term “atypical” has been removed.

  1. The term “abnormal clinical signs” (lines 92-93) should be defined. Do the authors consider mild neurological deficits to be abnormal in EHV-1 infections?

The term “abnormal” has been removed to only refer to clinical signs associated with disease.

  1. The Discussion could be reduced. For example, paragraph #2 (lines 204 to 248) rambles somewhat, discusses a gallimaufry of topics, and is overly long. 5. Line 232: “The present data is…”; Change “is” to “are”

The paragraph has been shortened as requested by the reviewer.

Reviewer 2 Report

The authors present a very thorough documentation of an outbreak and successful treatment of a novel EHV-1 genotype in an EHV-1-vaccinated population, which given the observed prevalence of neurological deficits and potential EHM is highly relevant. Apart from the basic data, the discussion section thoroughly describes clinical signs as well as vaccines and treatment options, which I appreciate.
One point of critique is the Conclusions section, which seems redundant as it is mostly just a summary of the observations and reads like a second Abstract. This section should state clearly and briefly the main conclusions and provide an explanation of the importance and relevance of the study to the field. I suggest removing lines 326 to 337, as this is a repetitive summary. The content of the remaining four lines may as well be integrated as a last section of the discussion. Apart form this, I only suggest a few minor corrections.

Minor comments:

Table 1
I suppose it would be a bit easier to read using “+” and “-“ symbols rather than ‘positive’ and ‘negative’ spelled out (and some other symbol for ‘doubtful’).
Also, instead of column 2 containing letters and temperatures, I suggest using four columns (for F, A, DLE, and ND) with a symbol if true (and empty cell if not), apart from F where simply the temperature could be shown. This would make these parameters easier to align with the qPCR and sero results.

Abstract 
line 17    perhaps add number of clinically infected horses after the percentage
line 19    change wording (“displayed viral load […] via qPCR” seems awkward).

Intro
line 45    “referred to as”
line 46    “referred to as”

Results
line 75    ‘tested’ rather than tests
line 80    ‘physiological’ rather than physical?
line 83    aged
line 89    temperatures

line 166/167    the abbreviation EHM was introduced above, remove the spelled-out term here

Methods
line 312    I suggest “Sequencing of an ORF 30 gene fragment” (rather than “partial sequencing”).
line 313    add a hyphen between 633 and bp

Author Response

Dear Reviewer,

We herewith submit the revisions to our manuscript entitled “Investigation of an EHV-1 Outbreak in the United States Caused by a new H752 Genotype” by Nicola Pusterla, Samantha Barnum, Julia Miller, Sarah Varnell, Barbara Dallap-Schaer , Helen Aceto and Aliza Simeon. We appreciate the time and comments provided by the reviewer, which we agree greatly improve the manuscript. We have addressed the suggested changes to the manuscript, and hope that our manuscript is now acceptable for publication.

Thank you for your consideration and we look forward to hearing back from you at your earliest convenience.

Sincerely,

Nicola Pusterla

Comments to Reviewer 2

One point of critique is the Conclusions section, which seems redundant as it is mostly just a summary of the observations and reads like a second Abstract. This section should state clearly and briefly the main conclusions and provide an explanation of the importance and relevance of the study to the field. I suggest removing lines 326 to 337, as this is a repetitive summary. The content of the remaining four lines may as well be integrated as a last section of the discussion. Apart form this, I only suggest a few minor corrections.

As mentioned by the reviewer, the conclusion has been changed to reflect the key observations gained from the investigation of the outbreak.

Table 1
I suppose it would be a bit easier to read using “+” and “-“ symbols rather than ‘positive’ and ‘negative’ spelled out (and some other symbol for ‘doubtful’).
Also, instead of column 2 containing letters and temperatures, I suggest using four columns (for F, A, DLE, and ND) with a symbol if true (and empty cell if not), apart from F where simply the temperature could be shown. This would make these parameters easier to align with the qPCR and sero results.

As suggested by the reviewer, the table has been changed to make it easier to read.

Abstract 
line 17    perhaps add number of clinically infected horses after the percentage

The number have been added as suggested.

line 19    change wording (“displayed viral load […] via qPCR” seems awkward).

Changed displayed with experienced.

Introduction
line 45    “referred to as”

Changed according to suggestion.

line 46    “referred to as”

Changed according to suggestion.

Results
line 75    ‘tested’ rather than tests

Changed according to suggestion.

line 80    ‘physiological’ rather than physical?

Changed according to suggestion.

line 83    aged

Changed according to suggestion.

line 89    temperatures

Changed according to suggestion.

line 166/167    the abbreviation EHM was introduced above, remove the spelled-out term here

Changed according to suggestion.

Methods
line 312    I suggest “Sequencing of an ORF 30 gene fragment” (rather than “partial sequencing”).

Changed according to suggestion.

line 313    add a hyphen between 633 and bp

Changed according to suggestion.